# Clinical and Histopathological Factors Influencing IgA Nephropathy Outcome

**DOI:** 10.3390/diagnostics11101764

**Published:** 2021-09-25

**Authors:** Andrzej Konieczny, Piotr Donizy, Tomasz Gołębiowski, Andrzej Tukiendorf, Agnieszka Hałoń, Mariusz Kusztal, Hanna Augustyniak-Bartosik, Magdalena Krajewska

**Affiliations:** 1Department of Nephrology and Transplantation Medicine, Wroclaw Medical University, 50-556 Wroclaw, Poland; andrzej.konieczny@umed.wroc.pl (A.K.); mariusz.kusztal@umed.wroc.pl (M.K.); hanna.augustyniak-bartosik@umed.wroc.pl (H.A.-B.); magdalena.krajewska@umed.wroc.pl (M.K.); 2Department of Pathomorphology and Oncological Cytology, Wroclaw Medical University, 50-556 Wroclaw, Poland; piotr.donizy@umed.wroc.pl (P.D.); agnieszka.halon@umed.wroc.pl (A.H.); 3Department of Public Health, Wroclaw Medical University, 50-367 Wroclaw, Poland; andrzej.tukiendorf@umed.wroc.pl

**Keywords:** IgA nephropathy, proteinuria, glomerular filtration rate

## Abstract

IgA nephropathy (IgAN) is the most frequent primary glomerulonephritis worldwide. Due to its heterogenicity, there is a need to establish robust biomarkers for IgAN, to support treatment decisions and evaluate the risk of progression to end-stage renal disease. Using both clinical and histopathological data, derived from renal biopsies, we aimed to find predictors of renal function deterioration and proteinuria reduction. Clinical and histopathological data of 80 patients with biopsy proven IgAN were analyzed. In a multivariate logarithmic regression model, the presence of endocapillary hypercellularity (E1) predicted a decline in estimated glomerular filtration rate (eGFR)of at least 50% with an odds ratio (OR) of 15.2, whereas serum albumin concentration had a negative influence on eGFR deterioration (OR 0.2). In the second multivariate model, the extent of interstitial fibrosis predicted the worsening of eGFR by 50% (OR 1.1) and serum albumin concentration had a protective impact (OR 0.1). In the univariate logarithmic regression, both the extent of interstitial fibrosis and the presence of endocapillary hypercellularity negatively correlated with the reduction in proteinuria below 1.0 g/24 h with an OR of 0.2 and 0.9, respectively. In our paper, we confirmed the utility of histopathological variables, especially endocapillary hypercellularity and interstitial fibrosis, and clinical parameters, particularly serum albumin concentration, in the prediction of both a decline in eGFR and a reduction in proteinuria in IgA nephropathy.

## 1. Introduction

IgA nephropathy (IgAN) is the most frequent primary glomerulonephritis worldwide [1], first described by Berger in 1968 [2]. Its diagnosis requires the demonstration of predominant mesangial IgA deposits on kidney biopsy [3]. The prevalence of IgAN varies across the world, with the highest values in East Asia, reaching 40% of biopsies in Japan, through 25% in Europe, 12% in the United States, and below 5% in Africa [4]. Studies with long-term follow-up found that IgAN is related to poor renal outcomes [5], and about 30–40% of IgAN patients progress to end-stage renal disease (ESRD) within 10–25 years [6]. IgAN is described by a highly variable clinical course from an entirely benign incidental condition, with the absence of proteinuria and solely erythrocyturia in sediment, to severe nephrotic syndrome and rapidly progressive kidney failure [1].

There is a need for establishing robust biomarkers for IgA nephropathy to aid with diagnosis, treatment decisions, and risk prediction for progressive disease [7].

The Oxford Classification of IgA Nephropathy (MEST score) offers an opportunity to use histology for the prediction of renal outcome, independently of proteinuria, blood pressure, and estimated glomerular filtration rate (eGFR) [8]. The scale includes mesangial hypercellularity(M), endocapillary hypercellularity (E), segmental sclerosis (S), interstitial fibrosis/tubular atrophy (T), and presence of crescents (C).The European Validation Study of the Oxford Classification of IgAN (VALIGA) confirmed the association of M1, S1, and T1/2 with renal outcomes, as well as the association of M1 and E1 with subsequent increase in proteinuria [9]. The combination of MEST score with blood pressure, proteinuria, and eGFR at the time of biopsy predicted the composite renal outcome similar to using clinical data over 2 years of follow-up [10].

Recently, a new risk-prediction tool, including eGFR, blood pressure, use of immunosuppression, angiotensin system blockers (RAS), and proteinuria at the time of biopsy, while also incorporating MEST score, was proposed. It accurately predicted the risk of a 50% decline in eGFR or development of ESKD up to 7 years after biopsy [11].

In this single-center, retrospective study, the clinical and pathological data of IgAN were analyzed to assess the predictability of both a decline in eGFR and a reduction in proteinuria.

## 2. Materials and Methods

### 2.1. Patients

Eighty adult patients from a single center, with biopsy-proven IgA nephropathy, were enrolled in this retrospective study, from 1 January 2012 to 31 December 2019. The exclusion criteria were secondary causes of IgAN; in our center, this included hepatic cirrhosis and Crohn’s disease.

The study was conducted according to the guidelines of the Declaration of Helsinki and approved by the Institutional Review Board of Wroclaw Medical University (KB 608/2019; 10 April 2019). 

### 2.2. Kidney Specimen Histology

Specimens obtained by renal biopsy were assessed by experienced pathologists (PD and AH), according to the Oxford classification of IgAN [12]. Additionally, the extent of interstitial fibrosis was assessed by evaluating the exact percentage of renal cortex fibrosis, and the intensity of interstitial inflammation was categorized on a semiquantitative scale from 0 to 3.

### 2.3. Clinical Parameters

The following clinical parameters were assessed: age, gender, blood pressure (systolic (SBP), diastolic (DBP), and mean (MAP) calculated as ((2 × DBP) + SBP)/3), erythrocyturia in urinary sediment, neutrophil-to-lymphocyte ratio (NLR), platelet-to-lymphocyte ratio (PLR), and serum concentrations of total cholesterol, triglycerides, uric acid, fasting glucose, serum creatinine, serum albumin, and total protein. The number of red blood cells (RBCs) in urinary sediment was calculated per high-power field (HPF). In the case of RBC/HPF ≤5, RBC/HPF of 6–20, and RBC/HPF >20, the extent of erythrocyturia was assessed as absent, moderate, and severe, respectively. The enzymatic method was used for urea, and Jaffe’s method was used for creatinine evaluation. Kidney function was expressed as eGFR using the MDRD equation [13]. Renal function decline (the slope of eGFR) was assessed between the biopsy time and the last control visit in the clinic before December 2019. On this basis, patients were divided into two groups: those who achieved and who did not achieve an eGFR reduction of more than 50%. The remaining parameters were evaluated using routine methods.

### 2.4. Clinical Management of the Patients

All patients, with proteinuria greater than 1.0 g per day, were treated with angiotensin-converting enzyme inhibitors (ACEI) or angiotensin receptor blockers ARB (in the case of ACEI intolerance) at maximum tolerated doses, to obtain blood pressure below 125/75 mmHg. We did not include the ACEI or ARB doses in our analysis due to their variability. Patients who did not achieve a reduction in proteinuria below 1.0 g per day, after 6 months of treatment with ACEI or ARB, received corticosteroids according to the scheme proposed by Pozzi et al. [14] (i.v. bolus injection of 1 g of methylprednisolone for 3 days at 1, 3, and 5 months each, followed by oral prednisone 0.5 mg/kg every other day for 6 months).

### 2.5. Statistical Analysis

Statistical analysis was performed using Statistica ver. 13 software (StatSoft, Tulsa, OK, USA). Quantitative continuous data were expressed as the mean and standard deviation (SD). The distribution was tested with the use of the Kolmogorov–Smirnov test. Differences between the two groups were assessed with the use of a *t*-test. Categorical variables were expressed as absolute (*N*) and percentage (%) values and compared using the χ^2^ test. To predict the values of decline in eGFR and reduction in proteinuria, logistic regression models were built. Firstly, univariate models were made to determine the best predictors. Then, multivariate regression was performed, including statistically significant parameters from univariate models. A *p*-value <0.05 was considered statistically significant.

## 3. Results

### 3.1. Patients’ Baseline Characteristics

Eighty patients with primary IgA nephropathy were enrolled in the study, among them 35 women (44%) and 45 men (56%), aged 39 ± 14 years. Detailed patients’ characteristics at baseline are presented in Table 1.

### 3.2. Histopathological Baseline Data

MEST-C classification, the extent of interstitial fibrosis, and the degree of interstitial inflammation are presented in Table 2.

Figure 1 presents examples of histopathological findings of IgAN.

### 3.3. Association between Clinical and Histopathologic Features and Decline in eGFR >50%

Seven patients (8.8%) suffered from a decline in eGFR of at least 50% during the observation period. Table 3 presents basic clinical characteristics and statistical differences between both groups. The follow-up time in both groups was comparable and amounted 35 ± 18 and 37 ± 20 months, *p* = 0.34. Urinary protein excretion was assessed as urine protein-to-creatinine ratio (UPCR). Patients suffering from a decline in eGFR by at least 50% had statistically significantly higher PLR and lower concentrations of both serum albumin and total protein. They also had higher levels of proteinuria as assessed using UPCR. Furthermore, they used prednisone more frequently before biopsy.

Renal biopsies in the group with eGFR deterioration showed a higher degree of endocapillary hypercellularity (E1) and extent of interstitial fibrosis. Differences between both groups in terms of histopathological parameters are presented in Table 4.

Using nonlinear estimation, six parameters among those presented in Table 1 and Table 2 were found to have a statistically significant impact on eGFR deterioration >50%. Data are presented in Table 5. Data without statistical significance are not presented.

Using the parameters selected by the univariate logistic regression model, we attempted to build a multivariate model, predicting 50% eGFR reduction. Only those consisting of two variables reached statistical significance. The best-fitting multivariate logistic regression models are presented in Table 6 and Table 7.

### 3.4. Association between Clinical and Histopathologic Features and Proteinuria Reduction below 1.0 g/24 h

In the analyzed group, 38 (47.5%) patients presented urinary protein excretion higher than 1.0 g (as estimated by UPCR) at the time of biopsy. In 23 (61%) of them, daily protein loss was reduced below 1.0 g. The mean time to proteinuria reduction was 8 ± 12 months. In Table 8 and Table 9, baseline parameters and differences between the groups are demonstrated.

Patients in whom reduction in proteinuria was achieved presented a lower extent (assessed as a percentage value) of interstitial fibrosis and lack of endocapillary hypercellularity, compared to those in whom no reduction in proteinuria was achieved.

A univariate logistic regression model was built (Table 10) on the basis of the data presented in Table 1 and Table 2. Data without statistical significance are not presented. Among clinical and histological parameters, endocapillary hypercellularity and interstitial fibrosis were found to have a negative impact on the reduction in protein loss below 1.0 g/24 h.

Only a univariate logistic regression model, including parameters influencing a decrease in proteinuria <1.0 g/24 h, was built. In contrary to 50% eGFR decline, the multivariate model did not reach statistical significance.

### 3.5. Correlation between Endocapillary Hypercellularity and Clinical Parameters

A comparison of clinical parameters between patients based on endocapillary hypercellularity is presented in Table 11.

None of the clinical parameters differed between the groups on the basis of endocapillary hypercellularity.

Patients with E1 presented higher mesangial hypercellularity, interstitial inflammation, and extent of interstitial fibrosis (presented in Table 12), while the eGFR decline was also more strongly influenced by the presence of E1 in the kidneys. Kaplan–Meier survival is presented in Figure 2.

## 4. Discussion

IgA nephropathy is the most prevalent primary glomerulonephritis, leading to ESRD; however, due to its heterogeneity, it is challenging to accurately identify patients at high risk of kidney function deterioration. In addition, it should be noted that not all patients may benefit from aggressive immunosuppressive therapy [15]. On this basis, the need for a tool to help predict the course of the disease and identify patients at risk of developing ESRD is extremely important. There is evidence that several clinical features at presentation may predict the risk of progression. The most important factors include the level of proteinuria, hypertension, and renal function at the biopsy [16]. The Oxford Classification of IgAN provides data derived from renal biopsy assessment, including the degree of mesangial hypercellularity (M), endocapillary hypercellularity (E), segmental sclerosis (S), interstitial fibrosis/tubular atrophy (T), and presence of crescents (C). In the VALIGA cohort, the value of the MEST score was verified to predict the degree of renal function deterioration, as well as survival without ESRD, or a 50% reduction in initial GFR [17].

In our study, a significant correlation was found between the deterioration of renal function, as expressed by a decline in eGFR by 50%, compared to the baseline values, and the level of proteinuria. The group with a 50% decrease in eGFR had significantly lower concentrations of both serum albumin (2.5 ± 0.6 g/dL vs. 3.6 ± 0.9 g/dL, *p* = 0.002) and total protein (4.8 ± 0.9 g/dL vs. 6.2 ± 1.2 g/dL, *p* = 0.004) at baseline. It was accompanied by more severe protein loss, as expressed by UPCR (3.6 ± 2.5 vs. 1.7 ± 2.1, *p* = 0.02). This finding is consistent with previous papers, pointing out the level of proteinuria as a crucial factor in the progression of renal insufficiency among IgAN patients [18].

Moreover, there was a correlation between the presence of endocapillary hypercellularity (E1) and the risk of eGFR deterioration. In both univariate and multivariate logistic regression models, the presence of E1 increased the odds ratio of eGFR decline. On the one hand, this finding is in accordance with a study published by Edstrom et al., showing that the presence of endocapillary hypercellularity was associated with poor renal outcome, expressed by end-stage renal disease or eGFR reduction >50% [19]. Similar findings were presented in a paper by Chakera et al., where, in the multivariate regression, baseline eGFR, proteinuria, and endocapillary hypercellularity were independent predictors of time to ESRD and a rapid decline in eGFR [20]. Reversely, conclusions based on an analysis of the VALIGA cohort confirmed the value of the MEST score in predicting the rate of renal function deterioration, but the E1 score did not predict any of the outcomes [17], including kidney function decline.

Platelet-to-lymphocyte ratio (PLR) calculated as platelet count divided by lymphocyte count has been established as a novel marker of malignancies, cardiovascular diseases, and chronic obstructive pulmonary disease [21,22,23]. In Kaplan–Meier analysis, PLR more than 137 was found to be a predictive factor of poor renal outcome ESRD [24]. Both NLR and PLR were assessed as markers of inflammatory states, and their prognostic significance has been established in several conditions such as rheumatoid arthritis, solid tumors, and coronary artery disease. NLR and PLT reflect the excess of inflammatory response, based on complement activations and the production of proinflammatory components such as C3a and C5a, responsible for the migration of macrophages and lymphocytes. The chronic inflammation localized within the glomeruli leads to the production of extracellular matrix components, consequently initiating progressive tubulointerstitial fibrosis [25].

In our work, PLR was shown, in univariate analysis, to have a predictive influence on eGFR deterioration.

It is also worth mentioning that patients with endocapillary lesions were more likely to receive immunosuppressive treatment. In our study, 18 patients were treated with prednisone already at the time of biopsy, and the number of these patients increased after the diagnosis. There are two studies in which no patient received immunosuppression, and both reported that E1 was independently associated with loss of renal function [20,26]. Among those who received no immunosuppression, the rate of renal function decline was 5.4 ± 11.1 mL/min/1.73 m^2^ per year in those with endocapillary lesions, compared with 2.6 ± 5.1 mL/min/1.73 m^2^ per year in those without endocapillary proliferation (*p* = 0.02) [21]. These studies suggest that the employment of immunosuppression may mask the predictive value of endocapillary hypercellularity in renal outcomes and is only useful in situations where immunosuppression was not applied [27]. In our paper, the number of patients receiving immunosuppression before biopsy did not differ in correlation to the degree of endocapillary hypercellularity.

Using multivariate analysis, we found an influence of E1 on eGFR decline, whereas serum albumin concentration had a protective impact. The second multivariate model involving serum albumin concentration and interstitial fibrosis had an impact on eGFR worsening.

In our study, we also found that the presence of E1 influences blood pressure control. Although the number of patients with diagnosed hypertension did not differ between group E1 and E0, we observed a statistically significant difference in systolic blood pressure. Patients with the presence of E1 demonstrated a mean SBP of 137 ± 18 mmHg, in contrast to those with E0 who presented a mean SBP of 129 ± 15 mmHg. The higher values of blood pressure might be related to more active kidney disease, stimulating hormonal disorders responsible for blood pressure control, including activation of the renin–angiotensin–aldosterone system.

It is worth mentioning that, although the presence of T lesions in our group did not reach statistical significance, both in the univariate and in the multivariate logistic regression models, we were able to prove that the extent of interstitial fibrosis, at the time of biopsy, influenced the deterioration of renal function. It is now a widely accepted paradigm that the degree of renal fibrosis correlates well with kidney function, both in native kidneys and in kidney grafts [28]. The explanation of this difference might be the fact that, in the Oxford classification of IgAN, a T0 score corresponds to tubular atrophy/interstitial fibrosis <25% [12]. In relation to the MEST-C Oxford classification of IgAN, all those values are within the T0 score. In our work, the mean extent of interstitial fibrosis was 6.6% ± 2.2% in the group without eGFR decline vs. 11.1% ± 3.2% in the group with 50% eGFR deterioration. Similarly, in patients in whom a reduction of proteinuria below 1.0 g/24 h was achieved, the extent of interstitial fibrosis reached 4.0% ± 5.3% vs. 12.9% ± 12.3% among those in whom no such reduction was achieved. To explain those discrepancies, future studies should be conducted with the use of digital pathology tools.

Since the level of proteinuria, especially exceeding 1.0 g per day, is a key factor of renal insufficiency progression [17], we focused on patients in whom daily urinary protein excretion fell below 1.0 g. Furthermore, when renal function deteriorated, we found a statistically significant correlation, in logistic regression, between E1 and the degree of interstitial fibrosis and the likelihood of reduced protein excretion in the urine. Both factors were negatively associated with the odds ratio of proteinuria reduction <1.0 g/24 h. A crucial key factor, connecting proteinuria and progression of kidney insufficiency, is the extent of interstitial fibrosis. It is known that the proteins passing the glomerular filtration barrier have the potential to initiate the migration of inflammatory cells (mainly macrophages), consequently inducing interstitial fibrosis [29].

The main limitation of this study was its small sample size. Although IgA nephropathy is the most common type of primary glomerulonephritis, it is generally a rare disease in the entire population. All available patients with biopsy-proven IgA nephropathy and at least 6 months of follow-up were included in the study, after excluding patients with secondary causes. The strength of the study is that we presented the results from one center that provides care for the large region of southwestern Poland and concerns an ethnically homogeneous group. Additionally, the same approach was used to treat all participants.

## 5. Conclusions

In our paper, we confirmed the utility of the MEST-C score, especially endocapillary hypercellularity, in the prediction of renal function decline, expressed as a 50% decrease in eGFR. A similar conclusion was made for a reduction in proteinuria below 1.0 g/24 h. In addition to histopathological findings, the level of proteinuria expressed by initial serum albumin concentration was found to have an impact on final kidney function.

## Figures and Tables

**Figure 1 diagnostics-11-01764-f001:**
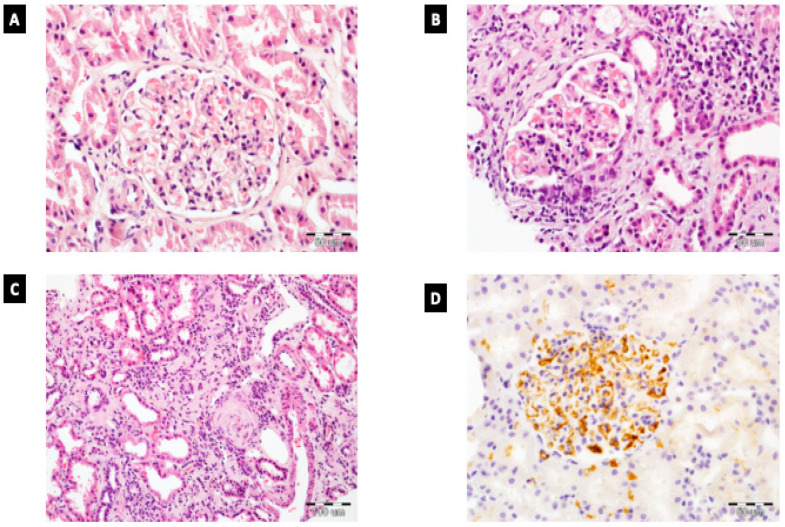
Histopathology of IgAN. (**A**) Mesangial hypercellularity without endocapillary hypercellularity in glomerulus in IgAN; HE, 400×. (**B**) Glomerulus with cellular crescent and endocapillary hypercellularity. Presence of moderate interstitial inflammation; HE, 400×. (**C**) Extensive tubular atrophy and interstitial fibrosis in IgAN patient; HE, 200×. (**D**) Enhanced IgA reactivity in glomerulus (evaluated by immunohistochemistry); 400×.

**Figure 2 diagnostics-11-01764-f002:**
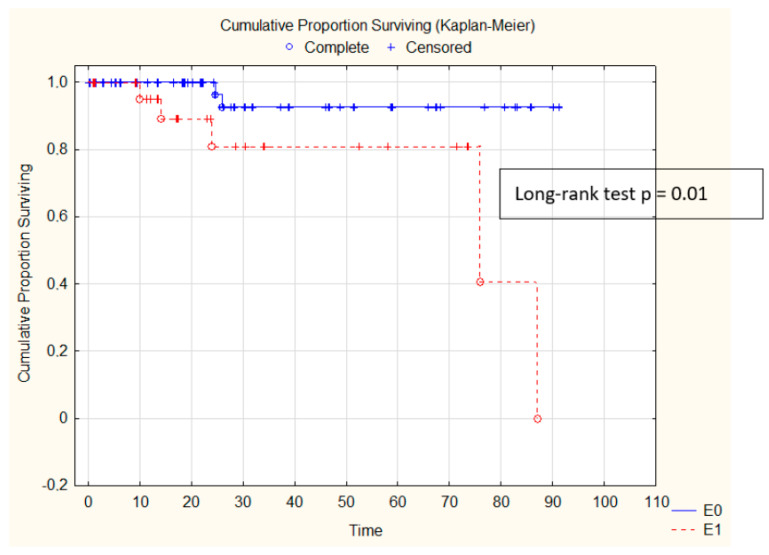
Kaplan–Meier eGFR decline analysis depending on endocapillary hypercellularity.

**Table 1 diagnostics-11-01764-t001:** Patients’ baseline clinical characteristics.

Variable	Baseline Values
Female/male	35/45 (44%/66%)
Age (years)	39 ± 14
SBP (mmHg)	131 ± 16
DBP (mmHg)	80 ± 11
MAP (mmHg)	95 ± 12
NLR	2.4 ± 1.5
PLR	140 ± 59
Total cholesterol (mg/dL)	258 ± 104
Triglycerides(mg/dL)	172 ± 91
Uric acid (mg/dL)	6.6 ± 1.7
Fasting glucose (mg/dL)	93 ± 11
Serum albumin (g/dL)	3.5 ± 0.9
Total protein (g/dL)	6.0 ± 1.2
Serum creatinine(mg/dL)	1.4 ± 1.4
eGFR (mL/min/1.73 m^2^)	68 ± 24
UPCR(g/g)	1.9 ± 1.6
Erythrocyturia—moderate	33 (41%)
Erythrocyturia—severe	28 (35%)
Hypertension	39 (49%)
ACEI treatment	52 (65%)
Prednisone use	18 (23%)

Abbreviations: SBP, systolic blood pressure; DBP, diastolic blood pressure; MAP, mean arterial blood pressure; NLR, neutrophil-to-lymphocyte ratio; PLR, platelet-to-lymphocyte ratio; eGFR, estimated glomerular filtration rate; UPCR, urine protein-to-creatinine ratio; ACEI, angiotensin-converting enzyme inhibitor.

**Table 2 diagnostics-11-01764-t002:** Histopathologic parameters of renal biopsies.

Variable	Values
M0	14 (17.5%)
M1	66 (82.5%)
E0	56 (70%)
E1	24 (30%)
S0	40 (50%)
S1	40 (50%)
T0	79 (98.75%)
T2	1 (1.25%)
C0	72 (90%)
C1	7 (8.75%)
C2	1 (1.25%)
Interstitial inflammation—0	35 (44%)
Interstitial inflammation—1	38 (48%)
Interstitial inflammation—2	7 (9%)
Interstitial fibrosis (%)	7 ± 8

Abbreviations: M, mesangial hypercellularity; E, endocapillary hypercellularity; S, segmental sclerosis; T, interstitial fibrosis/tubular atrophy; C, presence of crescents.

**Table 3 diagnostics-11-01764-t003:** Comparison between groups depending on decline in eGFR. The statistically significant parameters are bolded in the table.

Parameter	eGFR Reduction > 50%		*p*-Value
	No (*N* = 73)	Yes (*N* = 7)	
Age (years)	39 ± 13	44 ± 19	0.36
SBP (mmHg)	131 ± 16	133 ± 16	0.79
DBP (mmHg)	80 ± 11	79 ± 0	0.7
MAP (mmHg)	96 ± 12	95 ± 10	0.89
NLR	2.4 ± 1.5	3.0 ± 1.9	0.27
**PLR**	**135 ± 54**	**189 ± 86**	**0.02**
Total cholesterol (mg/dL)	256 ± 104	281 ± 105	0.55
Triglycerides (mg/dL)	171 ± 92	183 ± 95	0.74
Uric acid (mg/dL)	6.5 ± 1.8	7.0 ± 1.0	0.45
Fasting glucose (mg/dL)	93 ± 11	93 ± 12	0.93
**Serum albumin (g/dL)**	**3.6 ± 0.9**	**2.6 ± 0.6**	**0.002**
**Total protein (g/dL)**	**6.2 ± 1.2**	**4.8 ± 1.9**	**0.004**
Serum creatinine (mg/dL)	1.4 ± 1.1	1.1 ± 0.3	0.6
eGFR (mL/min/1.73 m^2^)	67 ± 24	81 ± 26	0.16
**UPCR (g/g)**	**1.7 ± 2.1**	**3.6 ± 2.5**	**0.02**
Erythrocyturia—moderate	29 (40%)	4 (57%)	0.62
Erythrocyturia—severe	27 (37%)	1 (14%)	0.49
Hypertension	33 (45%)	6 (86%)	0.1
ACEI treatment	47 (64%)	5 (71%)	0.97
**Prednisone use at the time of biopsy**	**13 (18%)**	**5 (71%)**	**0.006**
**Prednisone use after biopsy**	**32 (44%)**	**6 (86%)**	**0.08**

**Table 4 diagnostics-11-01764-t004:** Comparison of histopathological features between the groups, depending on a 50% deterioration of eGFR. The statistically significant parameters are bolded in the table.

MEST-C	eGFR Reduction >50%		*p*-Value
	No (*N* = 73)	Yes (*N* = 7)	
M1	59 (80%)	7 (100%)	0.24
**E1**	**19 (26%)**	**5 (71%)**	**0.03**
S1	38 (52%)	2 (29%)	0.39
T2	1 (1.4%)	0 (0%)	0.1
C1-2	7 (9.6%)	1 (14%)	0.7
Interstitial inflammation—1–2	39 (53%)	6 (86%)	0.21
**Interstitial fibrosis (%)**	**6.6 ± 2.2**	**11.1 ± 3.2**	**0.02**

**Table 5 diagnostics-11-01764-t005:** Univariate logistic regression. Variables correlated to a decrease of eGFR by at least 50%.

Variable	Estimate	OR	95% CI		*p*-Value
E1	2.0	7.1	1.2	40.8	0.03
Interstitial fibrosis (%)	0.1	1.05	0.98	1.1	0.018
PLR	0.01	1.01	1	1.02	0.0003
Serum albumin (g/dL)	−1.3	0.3	0.1	0.7	0.01
Total protein (g/dL)	−0.9	0.4	0.2	0.8	0.01
UPCR (g/g)	0.3	1.3	1.0	1.8	0.04

**Table 6 diagnostics-11-01764-t006:** Multivariate logistic regression (first model). Variables correlated to an eGFR reduction of at least 50%.

Variable	Estimate	OR	95% CI		*p*-Value
E1	2.7	15.2	1.7	13.6	0.02
Serum albumin (g/dL)	−1.7	0.2	0.05	0.6	0.008

**Table 7 diagnostics-11-01764-t007:** Multivariate logistic regression (second model). Variables correlated to an eGFR reduction of at least 50%.

Variable	Estimate	OR	95% CI		*p*-Value
Interstitial fibrosis (%)	0.13	1.1	1.03	1.27	0.02
Serum albumin (g/dL)	−1.98	0.1	0.03	0.56	0.006

**Table 8 diagnostics-11-01764-t008:** Baseline clinical variables in the groups based on reduction of protein excretion in urine < 1.0 g/24 h.

Parameter	Proteinuria Reduction < 1.0 g/day		*p*-Value
	Yes (*N* = 23)	No (*N* = 15)	
Age (years)	45 ± 16	41 ± 15	0.42
SBP (mmHg)	135 ± 15	129 ± 14	0.24
DBP (mmHg)	80 ±9	79 ±9	0.55
MAP (mmHg)	97 ± 10	94 ± 9	0.36
NLR	2.3 ± 1.4	2.6 ± 2.2	0.56
PLR	144 ± 59	141 ± 55	0.87
Total cholesterol (mg/dL)	327± 130	274 ± 114	0.2
Triglycerides (mg/dL)	198 ± 118	191± 98	0.84
Uric acid (mg/dL)	6.9 ± 2	6.8 ± 1.7	0.81
Fasting glucose (mg/dL)	95 ± 14	90 ± 8	0.81
Serum albumin (g/dL)	2.8 ± 0.9	3.0 ± 0.9	0.49
Total protein (g/dL)	5.2 ± 1	5.4 ± 1.5	0.67
Serum creatinine (mg/dL)	1.6 ± 2.5	1.3 ± 0.4	0.7
eGFR (mL/min/1.73 m^2^)	71 ± 28	62 ± 21	0.32
UPCR (g/g)	3.6 ± 2.5	3.2 ± 2	0.32
Erythrocyturia—moderate	10 (43%)	8 (53%)	0.55
Erythrocyturia—severe	7 (30%)	1 (7%)	0.18
Hypertension	13 (57%)	8 (54%)	0.85
ACEI treatment	14 (61%)	11 (73%)	0.66
Immunosuppression use at the time of biopsy	8 (35%)	4 (27%)	0.6
Immunosuppression use after biopsy	23 (100%)	15 (100%)	1.0

**Table 9 diagnostics-11-01764-t009:** Histopathologic variables depending on proteinuria reduction below 1.0 g/24 h. The statistically significant parameters are bolded in the table.

MEST-C	Proteinuria Reduction <1.0 g/day		*p*-Value Chi-Square Test
	Yes (*N* = 23)	No (*N* = 15)	
M1	18 (78%)	12 (80%)	0.9
**E1**	**5 (22%)**	**9 (60%)**	**0.03**
S1	10 (43%)	7 (47%)	0.9
T2	0 (0%)	1 (7%)	0.8
C1-2	2 (7.7%)	2 (13%)	0.4
Interstitial inflammation—1–2	13 (57%)	11 (73%)	0.48
**Interstitial fibrosis (%)**	**4.0 ± 5.3**	**12.9 ± 12.3**	**0.004**

**Table 10 diagnostics-11-01764-t010:** Univariate logistic regression. Variables correlated to a decrease in proteinuria of <1.0 g/24 h.

Variable	Estimate	OR	95% CI		*p*-Value
E1	−1.7	0.2	0.4	0.8	0.03
Interstitial fibrosis (%)	−0.16	0.9	0.75	0.96	0.01

**Table 11 diagnostics-11-01764-t011:** Clinical characteristics between groups based on endocapillary hypercellularity.

Parameter	Endocapillary Hypercellularity		*p*-Value
	E0 (*N* = 56)	E1 (*N* = 24)	
Age (years)	40 ± 14	37 ± 13	0.36
SBP (mmHg)	129 ± 15	137 ± 18	0.05
DBP (mmHg)	79 ± 11	82 ± 12	0.42
MAP (mmHg)	94 ± 11	98 ± 13	0.18
NLR	2.6 ± 1.7	2.1 ± 0.9	0.15
PLR	144 ± 59	141 ± 55	0.87
Total cholesterol (mg/dL)	259 ± 110	258 ± 89	0.97
Triglycerides (mg/dL)	172 ± 91	174 ± 92	0.91
Uric acid (mg/dL)	6.5 ± 1.8	6.9± 1.6	0.35
Fasting glucose (mg/dL)	93 ± 12	92 ± 10	0.87
Serum albumin (g/dL)	3.5 ± 0.96	3.5± 0.8	0.9
Total protein (g/dL)	6.1 ± 1.2	5.9± 1.1	0.48
Serum creatinine (mg/dL)	1.4 ± 1.6	1.3 ± 0.4	0.8
eGFR (mL/min/1.73 m^2^)	70 ± 26	65 ± 19	0.35
UPCR (g/g)	1.8 ± 2.4	1.9 ± 1.5	0.92
Erythrocyturia—moderate	22 (39%)	11 (46%)	0.57
Erythrocyturia—severe	20 (36%)	8 (34%)	0.84
Hypertension	25 (45%)	8 (58%)	0.26
ACEI treatment	35 (63%)	17 (71%)	0.47
Immunosuppression use at the time of biopsy	13 (23%)	5 (21%)	0.82
Immunosuppression use after biopsy	25 (45%)	13 (54%)	0.83

**Table 12 diagnostics-11-01764-t012:** Histopathological variables related to endocapillary hypercellularity. The statistically significant parameters are bolded in the table.

MEST-C	Endocapillary Hypercellularity		*p*-Value
	E0 (*N* = 56)	E1 (*N* = 24)	
**M1**	**42 (75%)**	**24 (100%)**	**0.02**
S1	25 (45%)	7 (63%)	0.14
T2	0 (0%)	1 (4%)	0.66
**C1–2**	**0 (0%)**	**8 (34%)**	**0.001**
**Interstitial inflammation—1–2**	**23 (41%)**	**22 (92%)**	**0.001**
**Interstitial fibrosis (%)**	**4.4 ± 5.1**	**13 ± 11**	**0.004**

## Data Availability

The data presented in this study are available in this article.

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
