# Peer review of "Clinical and Histopathological Factors Influencing IgA Nephropathy Outcome"

_diagnostics, 2021, doi:10.3390/diagnostics11101764_

Round 1

Reviewer 1 Report

This single-center study retrospectively observes clinical and histopathological data of biopsy-proven IgAN patients to find predictors of renal function deterioration and proteinuria reduction after therapy. The authors conclude that histopathological characteristics such as endocapillary hypercellularity and interstitial fibrosis, and clinical variables such as serum albumin concentration have an impact on the decline of eGFR and reduction of proteinuria. Validation of the Oxford classification in various IgAN cohorts is a hot topic in nephrology and since results have been quite heterogeneous, further studies in different cohorts with different immunosuppression backgrounds, baseline proteinuria, and eGFRs are definitely welcome. However, I have some important concerns regarding the presentation of the results.

Major comments

  1. Rapid eGFR reduction is commonly defined as a drop in eGFR of >5 ml/min/1.73m2. Why have the authors chosen a 50% reduction in eGFR as the primary endpoint? Please provide a reference.
  2. Table 5 & Table 10. Univariate logistic regression should specify all variables that were considered for analysis, especially those that are known prognostic factors for poor prognosis in IgAN (e.g. baseline eGFR, sex, age, etc)
  3. Tables 5, 6, and 7 should be made into one table so that it is easier for the reader to see and correlate with the text.
  4. Table 10. Multivariate logistic regression for prognostic variables should be performed.
  5. Although previous studies have shown mixed results regarding the MEST-C classification, most have shown a correlation of the M, S, T, C scores with GFR reduction especially in cohorts with significant proteinuria and low baseline GFR. The authors should discuss further why such relevance was not seen in this study.
  6. How can PLR be linked to IgAN, especially for the prediction of poor eGFR? Please elaborate in the discussion section.

Minor comments

  1. Materials & Methods, Patients: Name of Institution? Single-center? Multicenter?
  2. Line 65: secondary causes such as? Please elaborate.
  3. ACE inhibitors are more commonly abbreviated as ACEi or ACEI
  4. Lines 194-196: This information is a repetition of the introduction.

The paper should be checked in detail for minor grammatical errors. Below are some suggestions:

  1. Line 24: ‘In the univariate logarithmic regression, ‘
  2. Line 64 & 114 & 129: When starting a sentence with a number, the number must be written in full.
  3. Line 83: RBC/HPF ≤ 5
  4. Line 86: (the slope of eGFR)
  5. Line 100: 1, 3 and 5
  6. Line 132: amounted to 35… p=0.34, respectively <- should be taken out (redundancy)
  7. Line 134: higher PLR, and lower concentrations
  8. Line 135: levels of proteinuria
  9. Line 194: ESRD in _30-40%
  10. The wording in the paragraph of lines 216-227 is awkward.
  11. In general, there are several errors such as space bars left out, especially in the discussion section.

Author Response

Major comments:

  1. Rapid eGFR reduction is commonly defined as a drop in eGFR of >5 ml/min/1.73m2. Why have the authors chosen a 50% reduction in eGFR as the primary endpoint? Please provide a reference.

50% reduction of eGFR or doubling of serum creatinine concentration are commonly accepted and adopted endpoints while assessing the decline of renal function, especially regarding both primary and secondary glomerulonephritis. Such endpoint was chosen based on other multiple publications describing the IgA nephropathy progression, e. g. Barbour, S.J.; Coppo, R.; Zhang, H.; Liu, Z.H.; Suzuki, Y.; Matsuzaki, K.; Katafuchi, R.; Er, L.; Espino-Hernandez, G.; Kim, S.J., et al. Evaluating a New International Risk-Prediction Tool in IgA Nephropathy. JAMA Intern Med 2019, 179, 942-952, doi:10.1001/jamainternmed.2019.0600.

  1. Table 5 & Table 10. Univariate logistic regression should specify all variables that were considered for analysis, especially those that are known prognostic factors for poor prognosis in IgAN (e. g. baseline eGFR, sex, age, etc)

In Table 5 only data with statistically significant impact were presented, those which did not reach significance were not presented just to avoid “busy” table. An appropriate explanation has been added while introducing the Table 5.

The same situation applied to Table 10. All clinical and histopathological parameters were included while building the regression function but eventually only those which achieved statistical significance were presented, just to make the tables more convenient to read.

  1. Tables 5, 6, and 7 should be made into one table so that it is easier for the reader to see and correlate with the text.

Table 5 is presenting univariate logistic regression of clinical and both histopathological parameters with significant impact on eGFR decline, whether Table 6 and 7 include multivariate models built on the univariate one. Therefore, to facilitate the reading, we decided to present them separately. Moreover, Tables 6 and 7 regard to two multivariate models and therefore are also presented separately.

  1. Table 10. Multivariate logistic regression for prognostic variables should be performed.

Only univariate logistic regression model, including parameters influencing decrease of proteinuria < 1.0 g/24 h, was built. In contrary to 50% eGFR decline, the multivariate model did not reach statistical significance. An appropriate part has been added to the text.

  1. Although previous studies have shown mixed results regarding the MEST-C classification, most have shown a correlation of the M, S, T, C scores with GFR reduction especially in cohorts with significant proteinuria and low baseline GFR. The authors should discuss further why such relevance was not seen in this study.

This might be one of the limitations of the study. In our group at the beginning of the study the mean eGFR was 68 ml/min/ 1.73 m2 also the extent of proteinuria was at sub-nephrotic level with mean UPCR 1.9 g, therefore the group mentioned in the comment, with nephrotic proteinuria and low baseline eGFR, was represented by only few patients.

  1. How can PLR be linked to IgAN, especially for the prediction of poor eGFR? Please elaborate in the discussion section.

Following part has been added to “Discussion”

Both NLR and PLR were assessed as markers of inflammatory states and their prognostic significance has been establish in several conditions like rheumatoid arthritis, solid tumors, and coronary artery disease. NLR and PLT reflect the excess of inflammatory response, based on complement activations and production of pro-inflammatory components like C3a and C5a, responsible for migration of macrophages and lymphocytes. The localized within the glomeruli chronic inflammation leads to production of extracellular matrix components, being in consequence initiators of progressive tubulointerstitial fibrosis.

Minor comments

  1. Materials & Methods, Patients: Name of Institution? Single-center? Multicenter?

This was a single center study. This information was supplemented both in introduction and materials and methods. The name of institution in provided together with authors affiliations, it was omitted in text on purpose, allowing blind review.

  1. Line 65: secondary causes such as? Please elaborate.

All patients with secondary IgAN were excluded. In our center there were patients with cirrhosis and Crohn’s disease. Such information was added to “Patients” section in “Materials and Methods”. 

  1. ACE inhibitors are more commonly abbreviated as ACEi or ACEI

It was corrected.

  1. Lines 194-196: This information is a repetition of the introduction.

This part was corrected.

The paper should be checked in detail for minor grammatical errors. Below are some suggestions:

  1. Line 24: ‘In the univariate logarithmic regression, ‘
  2. Line 64 & 114 & 129: When starting a sentence with a number, the number must be written in full.
  3. Line 83: RBC/HPF ≤ 5
  4. Line 86: (the slope of eGFR)
  5. Line 100: 1, 3 and 5
  6. Line 132: amounted to 35… p=0.34, respectively <- should be taken out (redundancy)
  7. Line 134: higher PLR, and lower concentrations
  8. Line 135: levels of proteinuria
  9. Line 194: ESRD in _30-40%
  10. The wording in the paragraph of lines 216-227 is awkward.
  11. In general, there are several errors such as space bars left out, especially in the discussion section.

All comments were corrected.

Reviewer 2 Report

1. In this study, author confirm the utility of the MEST-C score, especially endocapillary hypercellularity, in the prediction of renal function decline. 

However, there does not seem to be a new finding or novelty in this study, because, It was already known that endocapillary hypercellularity is very important for prognosis factors in IgAN (PLoS One 2019; 14(3): e0214414, J Nephrol. 2016 Jun;29(3):367-375. ). It also has been well known that MEST is associated with increased proteinuria and decreased glomerular filtration rate in IgAN for various study and analysis (Kidney Int. 2016 Jan;89(1):167-75. , Kidney Int. 2014 Oct;86(4):828-36., Clin J Am Soc Nephrol. 2011 Sep;6(9):2175-84 ., J Pediatr (Rio J). 2017 Jul-Aug;93(4):389-397., Kidney Int. 2017 May;91(5):1014-1021.). 

Author suggests strengths of the authors' study are the single center, the large region patients of south-western Poland, ethnically homogeneous group and the same approach for treatment of all participants. 

However, even considering author’s argument, the number of patients is too small, and in addition, It is a retrospective study.

2. The definition of a reduction of more than 50% in eGFR is unclear. It is necessary to confirm how long does it take to decrease in eGFR >50%, for example eGFR>50 / 1 year. If a decrease of more than 50% for the entire follow-up period, the follow-up period will not be the same for each patient, so the rate of decrease for eGFR should be defined and analyzed again.

3. Logistic regression was performed on the group with and without eGFR decrease by >50, but since there were only 7 patients with the eGFR decrease by  >50, It is difficult to trust the results. I do not know whether it is possible to analyze using the propensity score.  if it is possible, it need to analyze using propensity score. However,  the number of patients in the two groups is too small, and imbalance of number of patients between 2 groups is severe. More patient enrollment and additional analysis are needed.

Author Response

  1. In this study, author confirm the utility of the MEST-C score, especially endocapillary hypercellularity, in the prediction of renal function decline.However, there does not seem to be a new finding or novelty in this study, because, It was already known that endocapillary hypercellularity is very important for prognosis factors in IgAN (PLoS One 2019; 14(3): e0214414, J Nephrol. 2016 Jun;29(3):367-375.). It also has been well known that MEST is associated with increased proteinuria and decreased glomerular filtration rate in IgAN for various study and analysis (Kidney Int. 2016 Jan;89(1):167-75., Kidney Int. 2014 Oct;86(4):828-36., Clin J Am Soc Nephrol. 2011 Sep;6(9):2175-84., J Pediatr (Rio J). 2017 Jul-Aug;93(4):389-397., Kidney Int. 2017 May;91(5):1014-1021.). 

In our study we have shown the importance of endocapillary hypercellularity in both predicting the renal function decline and diminishing the extent of proteinuria, in IgA nephropathy. On one hand VALIGA study did not proved the utility of endocapillary hypercellularity as a predictor of IgAN outcome, but on the other hand papers by Edstrom et al. and Chakera et al. are in contrary to those findings. With the results of our work, we are in accordance to above cited findings even if the results are not revolutionary.

  1. Author suggests strengths of the authors' study are the single center, the large region patients of south-western Poland, ethnically homogeneous group and the same approach for treatment of all participants. 

Our group consisted of very homogenous population and therefore the therapeutic approach was similar to all of the patients.

  1. However, even considering author’s argument, the number of patients is too small, and in addition, It is a retrospective study.

The number of patients included to the study is a limitation, what we stressed in discussion. On the other hand, as being single nephrological center in south-west Poland, we included all eligible patients.

  1. The definition of a reduction of more than 50% in eGFR is unclear. It is necessary to confirm how long does it take to decrease in eGFR >50%, for example eGFR>50 / 1 year. If a decrease of more than 50% for the entire follow-up period, the follow-up period will not be the same for each patient, so the rate of decrease for eGFR should be defined and analyzed again.

The eGFR reduction of 50% was based on the difference between eGFR at baseline and the moment of its decline of 50%. The follow-up time in both groups were comparable and amounted 35±18 and 37±20 months, p= 0.34.

  1. Logistic regression was performed on the group with and without eGFR decrease by >50, but since there were only 7 patients with the eGFR decrease by>50, It is difficult to trust the results. I do not know whether it is possible to analyze using the propensity score. if it is possible, it need to analyze using propensity score. However, the number of patients in the two groups is too small, and imbalance of number of patients between 2 groups is severe. More patient enrollment and additional analysis are needed.

The number of included patients is the strongest limitation of presented study. During observation only 7 patients reached eGFR reduction. While statistical analysis the imbalance between the groups were considered.

Reviewer 3 Report

Reviewer Comments

The authors in the research article entitled " Clinical and Histopathological Factors Influencing IgA Nephropathy Outcomes" confirmed the utility of histopathological variables, especially endocapillary hypercellularity and interstitial fibrosis, and clinical parameters, in particular serum albumin concentration, in the prediction of both declines in eGFR and reduction in proteinuria in IgA nephropathy. The authors should consider the following points to improvise the article.

  1. A flowchart showing the methodology and overall representation of the manuscript.
  2. What were the inclusion criteria for recruiting patients? 
  3. Exclusion criteria were secondary causes of IgAN. What were the secondary causes of IgAN. List it ou.
  4. The authors should provide renal biopsies images showing interstitial inflammation and fibrosis. 
  5. There are too many tables in the main manuscript. Some of these tables can be converted into graphs or should be moved into supplementary data.
  6. The significant parameters in the table should be highlighted in bold to provide a clear representation.
  7. In this study, the patient sample size is too small which is the limitation of the study.

Author Response

The authors in the research article entitled " Clinical and Histopathological Factors Influencing IgA Nephropathy Outcomes" confirmed the utility of histopathological variables, especially endocapillary hypercellularity and interstitial fibrosis, and clinical parameters, in particular serum albumin concentration, in the prediction of both declines in eGFR and reduction in proteinuria in IgA nephropathy. The authors should consider the following points to improvise the article.

  1. What were the inclusion criteria for recruiting patients? 

All patient with de novo diagnosed, confirmed in biopsy, IgA nephropathy were included into study.

  1. Exclusion criteria were secondary causes of IgAN. What were the secondary causes of IgAN. List it out.

Secondary IgAN were liver cirrhosis and Crohn’s disease. This part was added to “Patients” section.

  1. The authors should provide renal biopsies images showing interstitial inflammation and fibrosis. 

The renal biopsies images were provided and added to the paper.

  1. There are too many tables in the main manuscript. Some of these tables can be converted into graphs or should be moved into supplementary data.

The tables will be moved to supplementary material.

  1. The significant parameters in the table should be highlighted in bold to provide a clear representation.

The statistically significant parameters were highlighted.

  1. In this study, the patient sample size is too small which is the limitation of the study.

We admit that the number of patients is a main limitation. We included all eligible patients from one center. On the other hand, the group is very heterogenous and with similar therapeutic approach, helping avoid the bias from previous studies.

Round 2

Reviewer 2 Report

No further comments. 

Reviewer 3 Report

The authors have addressed all the comments to improvise the manuscript.